# The Efficacy of Hydrogen Peroxide in Mitigating Cyanobacterial Blooms and Altering Microbial Communities across Four Lakes in NY, USA

**DOI:** 10.3390/toxins12070428

**Published:** 2020-06-29

**Authors:** Mark W. Lusty, Christopher J. Gobler

**Affiliations:** School of Marine and Atmospheric Sciences, Stony Brook University, Southampton, NY 11968, USA; mark.lusty@stonybrook.edu

**Keywords:** cyanobacteria, hydrogen peroxide, 16S rRNA, harmful algal blooms

## Abstract

Hydrogen peroxide (H_2_O_2_) has been proposed as an agent to mitigate toxic cyanobacterial blooms due to the heightened sensitivity of cyanobacteria to reactive oxygen species relative to eukaryotic organisms. Here, experiments were conducted using water from four diverse, eutrophic lake ecosystems to study the effects of H_2_O_2_ on cyanobacteria and non-target members of the microbial community. H_2_O_2_ was administered at 4 µg L^−1^ and a combination of fluorometry, microscopy, flow cytometry, and high throughput DNA sequencing were used to quantify the effects on eukaryotic and prokaryotic plankton communities. The addition of H_2_O_2_ resulted in a significant reduction in cyanobacteria levels in nearly all experiments (10 of 11), reducing their relative abundance from, on average, 85% to 29% of the total phytoplankton community with *Planktothrix* being highly sensitive, *Microcystis* being moderately sensitive, and *Cylindrospermopsis* being most resistant. Concurrently, eukaryotic algal levels increased in 75% of experiments. The bacterial phyla *Actinobacteria*, cyanobacteria, *Planctomycetes*, and *Verrucomicrobia* were most negatively impacted by H_2_O_2_, with *Actinobacteria* being the most sensitive. The ability of H_2_O_2_ to reduce, but not fully eliminate, cyanobacteria from the eutrophic water bodies studied here suggests it may not be an ideal mitigation approach in high biomass ecosystems.

## 1. Introduction

Cyanobacteria, or blue-green algae, are photosynthetic prokaryotes that are ubiquitous in fresh and marine waterbodies. Blooms of cyanobacteria in eutrophic waters can be associated with light attenuation and hypoxia, and some bloom-forming cyanobacteria are capable of producing a suite of toxins, most commonly the hepatotoxin, microcystin [1]. Consequently, the World Health Organization (WHO) and US EPA have set drinking water and bathing guidance values microcystin [2,3]. In addition to exposure through drinking water and bathing, cyanotoxins can be ingested through the consumption of fish and shellfish [4,5,6]. These toxins can also affect animals; between 2007 and 2011, there were 67 cases of dog poisonings due to toxic cyanobacteria blooms across the U.S., 38 of which were fatal [7].

The occurrence of harmful cyanobacterial blooms is often linked to excessive anthropogenic eutrophication [5,8,9] but reducing nutrient loads can be a difficult and lengthy process that can involve changing fertilizer and wastewater disposal practices. Hence, there is interest in identifying mitigation approaches that can selectively target and remove toxic cyanobacterial blooms in order to prevent exposure and harm. Hydrogen peroxide (H_2_O_2_) has been considered for this role [10,11,12]. As a strong oxidant, it is known for its disinfectant capabilities, is a naturally occurring compound in aquatic systems, and quickly decomposes into water and gaseous oxygen [13]. As H_2_O_2_ decomposes, it releases hydroxyl radicals, strong reactive oxygen species known to damage cells and inhibit photosynthetic activity by causing damage to photosystem II [14,15]. H_2_O_2_ has been shown to be specifically detrimental to the growth and function of cyanobacteria and capable of reducing biomass of *Microcystis* and *Planktothrix* by 50% in less than 48 h [10,11]. Cyanobacteria are known to be more sensitive to H_2_O_2_ than eukaryotic primary producers [13,14,16], and a previous study found *Microcystis aeruginosa* to be ten-times more sensitive than species of green algae and diatoms [13]. This may be due, in part, to the photosystems of cyanobacteria not being protected within an organelle [14]. In addition, unlike cyanobacteria, eukaryotic phytoplankton commonly produce enzymes such as ascorbate peroxidase that break down H_2_O_2_ and protect them from damage by reactive oxygen species (ROS) such as hydroxyl radicals [17]. A whole lake study examining mesozooplankton abundances, mostly *Daphnia* and *Diaphanosoma*, found that they were unaffected at 2 mg H_2_O_2_ L^−1^, a concentration that inhibited the cyanobacterium *Planktothrix* [11].

The effects of hydrogen peroxide on cyanobacteria has been well-documented in laboratory cultures [13,18,19]. However, research assessing the effect of H_2_O_2_ on other important members of planktonic communities such as picocyanobacteria, eukaryotes, and heterotrophic bacteria has been limited. It is important that effects of H_2_O_2_ on cyanobacteria and the rest of the prokaryotic and eukaryotic community are understood before H_2_O_2_ is widely used for mitigation purposes in natural ecosystems.

This project, therefore, sought to understand the effects of H_2_O_2_ on multiple genera of toxin-producing cyanobacteria (i.e., *Microcystis*, *Dolichospermum*, *Cylindrospermopsis*, and *Planktothrix*) as well as co-occurring plankton including picocyanobacteria, eukaryotic algae, and heterotrophic bacteria. This was done through a series of incubation experiments performed using environmental samples from four contrasting water bodies across Long Island, NY, USA. Microbial communities were assessed using standard (microscopy, fluorometry) and molecular (high throughput amplicon sequencing) approaches to establish a comprehensive assessment of the efficacy of H_2_O_2_ as a mitigation approach for toxic cyanobacterial blooms.

## 2. Results

### 2.1. Fluorometric Response of the Phytoplankton Community

During three experiments utilizing water from Lake Agawam, initial cyanobacterial biomass ranged from 73 to 243 µg Chla L^−1^, dominated by mixtures of *Microcystis*, *Planktothrix*, and *Dolichospermum*. Cyanobacterial biomass was significantly lower four to six days after exposure to 4 mg H_2_O_2_ L^−1^ in two of three experiments, with concentrations 52% (*p* < 0.001; Figure 1a) and 43% (*p* < 0.005; Figure 1c) lower than the control. Cyanobacterial biomass was reduced below the New York State Department of Environmental Conservation (NYSDEC) level of concern of 25 µg Chla L^−1^ in the one experiment that had the lowest initial concentration of these experiments (73 µg Chla L^−1^; Figure 1a). Initial green algal biomass was low (0–0.24 µg Chla L^−1^) but was significantly higher than the control following exposure to H_2_O_2_ in one of the experiments (*p* < 0.001; Figure 1a). Unicellular brown algae were fluorometrically undetectable during the Lake Agawam experiments.

Initial cyanobacterial biomass for the three Mill Pond experiments ranged from 45 to 366 µg Chla L^−1^, dominated by *Microcystis* and *Cylindrospermopsis*. Four or seven days after exposure to H_2_O_2_, cyanobacterial biomass was significantly lower than the control in all three experiments by 99% (*p* < 0.001; Figure 1d), 93% (*p* < 0.001; Figure 1e), and 95% (*p* < 0.001; Figure 1f), respectively, and below the level of concern of 25 µg Chla L^−1^ in two experiments (Figure 1d,f). Initial biomass of green algae in Mill Pond ranged from below detection to 7.3 µg Chla L^−1^ and rose higher than the control to 76 ± 7 µg Chla L^−1^ (*p* < 0.001; Figure 1d), 30 ± 2 µg Chla L^−1^ (*p* < 0.001; Figure 1e), and 342 ± 18 µg Chla L^−1^ (*p* < 0.001; Figure 1f), respectively, in all three experiments following H_2_O_2_ exposure. Initial unicellular brown algae biomass levels were below detection at the start of the three experiments, but values rose significantly to 5 ± 1 µg Chla L^−1^ (*p* < 0.005) and 24 ± 3 µg Chla L^−1^ (*p* < 0.001) in two of the three experiments, but remained undetectable in the third experiment (Figure 1d,f).

For the Georgica Pond experiment, initial cyanobacterial biomass was 58 µg Chla L^−1^ being dominated by *Aphanizomenon* (Figure 1g). Four days after treatment with 4 mg H_2_O_2_ L^−1^, cyanobacterial biomass was 99.8% lower than the control and nearly 0 µg Chla L^−1^ (*p* < 0.001; Figure 1g). Initial green algal biomass was 11 µg Chla L^−1^ and was 144% higher than the control following exposure to H_2_O_2_ (*p* < 0.01; Figure 1g). Initial unicellular brown algae biomass was 39 µg Chla L^−1^ and rose to 47 ± 13 µg Chla L^−1^ following H_2_O_2_ exposure, significantly higher than the control where concentration had fallen (*p* < 0.05; Figure 1g).

For the Roth Pond experiment, initial cyanobacterial biomass was 222 µg Chla L^−1^ and was dominated by *Microcystis* and *Cylindrospermopsis* (Figure 1h). Cyanobacterial biomass was 82% lower than the control (*p* < 0.001) at 19 ± 3 µg Chla L^−1^ in the 4 mg H_2_O_2_ L^−1^ treatment after six days. Initial green algae biomass was 29 µg Chla L^−1^ but rose in the H_2_O_2_ treatment to 403 ± 10 µg Chla L^−1^, 400% higher than the control (*p* < 0.001; Figure 1h). Unicellular brown algae were undetectable at the start of the experiment but were significantly higher than the control at 20 ± 3 µg Chla L^−1^ 6 days after H_2_O_2_ exposure (*p* < 0.005; Figure 1h).

In summary, during the eight incubation experiments among four locations assessed fluorometrically, cyanobacterial levels were significantly lowered following the addition of H_2_O_2_ than the control in seven experiments, green algae levels became significantly higher in H_2_O_2_ treatments relative to the control in six of eight experiments, and unicellular brown algae became significantly higher in H_2_O_2_ treatments relative to the control in four of eight experiments.

### 2.2. Detailed Assessment of Planktonic Responses to H_2_O_2_

Given the strong and significant effects of H_2_O_2_ on plankton communities during this first set of experiments, three additional experiments were performed utilizing water from three ecosystems with additional analyses performed to more fully assess the response of plankton communities to H_2_O_2_. In the first of these experiments from Lake Agawam, the initial cyanobacterial biomass was 103 µg Chl*a* L^−1^ and five days following exposure to 4 mg H_2_O_2_ L^−1^ was significantly lower than the control at 23 ± 2 µg Chl*a* L^−1^ (*p <* 0.001; Figure 2a). Initial green algae biomass concentration was undetectable but rose to 69 ± 4 µg Chl*a* L^−1^ after treatment with H_2_O_2_, a level significantly higher than the control (*p <* 0.001; Figure 2a). Unicellular brown algae biomass was undetectable at the start of the experiment but rose to levels significantly higher than the control at 0.4 ± 0.2 µg Chl*a* L^−1^ in the H_2_O_2_ treatment (*p <* 0.05; Figure 2a). Picocyanobacteria (*Cyanobium*) concentrations in Lake Agawam were initially 5,340 ± 360 cells mL^−1^ and decreased in the control to 1690 ± 80 cells mL^−1^ (Figure 2b). The H_2_O_2_ treatment was reduced by less, and was 110% higher relative to the control with a final concentration of 3,560 ± 90 cells mL^−1^ (*p <* 0.001; Figure 2b). Eukaryotic algae concentrations were 193% higher in the H_2_O_2_ treatment than the control at 9,090 ± 150 cells mL^−1^ (*p <* 0.001; Figure 2b). The initial concentration of heterotrophic bacteria was 7.0 × 10^5^ cells mL^−1^ and was 44% lower than the control after the addition of H_2_O_2_ at 3.45 × 10^5^ cells mL^−1^ (*p <* 0.001; Figure 2b). Diatom densities in Lake Agawam were 34 cells mL^−1^ and levels rose to be significantly higher in the treatment (*p <* 0.05) relative to the control to 120 ± 28 cells mL^−1^ (Figure 2c). Green algae concentrations were initially 657 cells mL^−1^ and were nearly six-fold higher in the treatment compared to the control at 3,700 ± 50 cells mL^−1^ (*p <* 0.001; Figure 2c). Initial *Microcystis* concentrations were 222 colonies mL^−1^ and were significantly reduced by H_2_O_2_ to below the control to 34 ± 7 colonies mL^−1^ (*p <* 0.001; Figure 2c). There were 76 *Dolichospermum* chains mL^−1^ at the start of the experiment and concentrations sharply declined to 4 ± 4 chains mL^−1^ following H_2_O_2_ addition, a level significantly lower than the control (*p <* 0.005; Figure 2c). Finally, following H_2_O_2_ addition *Planktothrix* concentrations were 50% of the control at 439 ± 43 chains mL^−1^ (*p <* 0.01; Figure 2c).

High throughput sequencing of the 16S rDNA gene indicated that the relative abundance of *Actinobacteria* in Lake Agawam was initially 17 ± 1% and dropped to 5 ± 1% five days after exposure to 4 mg H_2_O_2_ L^−1^, significantly lower than the control (*p <* 0.001). *Planctomycetes* was 4 ± 1% initially and was significantly reduced to 2 ± 1%, significantly lower than the control (*p <* 0.001), while *Verrucomicrobia* was 4 ± 1% and was reduced to 0.4 ± 0.1%, significantly lower than the control (*p <* 0.001; Figure 2d). The sequenced relative abundance of *Bacteroidetes* in Lake Agawam was significantly higher than the control after the addition of H_2_O_2_ at 42 ± 4% compared to an initial of 32 ± 1% (*p <* 0.001; Figure 2d). The sequenced relative abundance of *Proteobacteria* and other less abundant taxa abundances were not significantly altered by H_2_O_2_ (Figure 2d).

Among all prokaryotes, the sequenced relative abundance of cyanobacteria was initially 25 ± 2% but was reduced to lower than the control to 2 ± 1% after H_2_O_2_ exposure (*p <* 0.001). Among the cyanobacteria, *Planktothrix* was the most abundant taxa in the Lake Agawam experiment with an initial relative abundance of 94 ± 1% that was significantly lower than the control at 17 ± 2% following the addition of H_2_O_2_ (*p <* 0.001; Figure 2e). In contrast, *Microcystis* had an initial abundance of 4 ± 1% that was higher relative to the control at 68 ± 6% in the H_2_O_2_ treatment (*p <* 0.001; Figure 2e).

*Cylindrospermopsis* made up only 0.2 ± 0.1% of initial cyanobacterial sequences but was higher than the control at 6 ± 6% following the H_2_O_2_ addition (*p <* 0.001; Figure 2e). Multiplying cyanobacterial Chl*a* values by sequenced relative abundances provided an estimate of individual biomasses and revealed that *Planktothrix* biomass decreased significantly (*p* < 0.001), while *Microcystis* (*p* < 0.001) and *Cylindrospermopsis* (*p* < 0.001) biomasses increased (Figure 3).

For the experiment that provided a detailed assessment from the Mill Pond planktonic community, initial cyanobacterial biomass was 394 µg Chl*a* L^−1^ and was 96% lower than the control after five days following exposure to H_2_O_2_ at 13 ± 3 µg Chl*a* L^−1^ (*p <* 0.001; Figure 4a). Green algae were below detectable levels in Mill Pond at the start of the experiment but were significantly higher at 60 ± 7 µg Chl*a* L^−1^ (*p <* 0.001) following H_2_O_2_ treatment; unicellular brown algae were fluorometrically undetectable in this experiment (Figure 4a). Initial eukaryotic algae concentrations were 1.14 ± 0.30 × 10^4^ cells mL^−1^ and were unchanged by H_2_O_2_ but were 52% lower relative to the control (*p <* 0.001; Figure 4b).

*Cyanobium* and heterotrophic bacteria concentrations in the H_2_O_2_ treatment were not significantly different from the control (Figure 4b). Levels of diatoms were low in Mill Pond prior to the experiment (< 10 cells mL^−1^) but increased and were significantly higher than the control after the H_2_O_2_ addition with a final concentration of 157 ± 35 cells mL^−1^ (*p <* 0.01; Figure 4c). Green algae cell densities were initially 1290 ± 140 cells mL^−1^ and were significantly higher in the H_2_O_2_ treatment compared to the control at 2.65 ± 0.37 × 10^4^ cells mL^−1^ (*p <* 0.01; Figure 4c). Initial *Microcystis* concentrations were 108 ± 4 colonies mL^−1^ and were significantly lower than the control following exposure to H_2_O_2_ (*p <* 0.05; Figure 4c). *Cylindrospermopsis* initial concentration was 2.45 ± 0.14 × 10^5^ chains mL^−1^ and significantly decreased by 100% following exposure to H_2_O_2_ (*p <* 0.001; Figure 4c) while *Dolichospermum* and *Planktothrix* levels were unaffected.

High throughput sequencing revealed that H_2_O_2_ caused a significant decline in the sequenced relative abundance of several bacterial groups in Mill Pond including *Actinobacteria*, *Planctomycetes*, and *Verrucomicrobia* (*p <* 0.005; Figure 4d). In contrast, the sequenced relative abundances of *Bacteroidetes* and *Proteobacteria* were higher in the H_2_O_2_ treatment relative to the control (*p <* 0.001; Figure 4d). Among cyanobacteria identified via sequencing of 16S rDNA, *Cylindrospermopsis* was the dominant operational taxonomic unit (OTU) (88 ± 1% of cyanobacteria sequences) in initial sequences and increased in relative abundance following H_2_O_2_ addition to 98 ± 1%, significantly higher than the control (*p <* 0.001). In contrast, the sequenced relative abundances of *Microcystis* and *Nodosilinea* were significantly lowered by H_2_O_2_ relative to the control (*p <* 0.001; Figure 4e). Estimated changes in absolute abundances based on fluorometry and sequencing revealed that, despite the differential sensitivities of differing cyanobacterial groups to H_2_O_2_, the biomass of *Cylindrospermopsis* (*p* < 0.001), *Microcystis* (*p* < 0.01), and *Nodosilinea* (*p* < 0.001) all significantly declined in the treatment relative to the initial levels and the control (Figure 5).

Finally, during the Roth Pond experiment, initial cyanobacteria biomass was 59 µg Chl*a* L^−1^ and was significantly lower in H_2_O_2_ treatments at 39 ± 3 µg Chl*a* L^−1^ relative to the control seven days after exposure (*p <* 0.01; Figure 6a). Initial green algae biomass was 141 µg Chl*a* L^−1^ but was not significantly altered by H_2_O_2_, while unicellular brown algal biomass levels were 100% higher in the H_2_O_2_ treatment compared to the control at 239 ± 12 µg Chl*a* L^−1^ (*p <* 0.01; Figure 6a). Flow cytometrically quantified *Cyanobium* concentrations in the Roth Pond were 99.8% lower in the H_2_O_2_ treatment compared to the control (*p <* 0.001) whereas levels of eukaryotic algae were unchanged (Figure 6b). Levels of heterotrophic bacteria were 33% higher than the controls seven days after the addition of H_2_O_2_ (*p <* 0.005; Figure 6b). Green algae identified microscopically were 111% higher in the H_2_O_2_ treatment compared to the control at 1.28 ± 0.06 × 10^5^ cells mL^−1^ (*p <* 0.001; Figure 6c), while diatom levels were unchanged (Figure 6c). *Microcystis* concentrations were 1320 ± 180 colonies mL^−1^ and were reduced by 100% by H_2_O_2_ (*p <* 0.001) while *Dolichospermum* concentrations were unchanged (Figure 6c). Final, total concentration of microcystin was 0.56 ± 0.13 µg L^−1^ in the control, and significantly higher at 0.85 ± 0.06 µg L^−1^ in the H_2_O_2_ treatment (*p <* 0.005).

Sequencing of the 16S rDNA indicated *Actinobacteria* were significantly higher in the H_2_O_2_ treatment compared to the control (*p <* 0.001) whereas cyanobacteria and *Verrucomicrobia* relative abundances were significantly lower than the control (*p <* 0.001; Figure 6e). Among the cyanobacteria, *Microcystis* and *Cyanobium* were the two most abundant genera in Roth Pond and the relative abundance of both was lower in the H_2_O_2_ treatment compared to the control (*p <* 0.001; Figure 6e). In contrast, *Cylindrospermopsis* relative abundance was initially 4 ± 1% and increased to 28 ± 9% following in the H_2_O_2_ treatment and was significantly higher than the control (*p <* 0.001). Despite the lower relative and absolute abundances of *Microcystis* during this experiment, the concentration of the cyanotoxin, microcystin, was marginally higher in the H_2_O_2_ treatment than the control, at 0.9 ± 0.1 µg L^−1^ compared to 0.6 ± 0.1 µg L^−1^ (*p <* 0.05). The individual biomasses (relative abundance multiplied by cyanobacterial Chl*a*) for *Microcystis* (*p* < 0.05) and *Cyanobium* (*p* < 0.005) significantly declined in response to H_2_O_2_, while total biomass of *Planktothrix* (*p* < 0.01) and *Cylindrospermopsis* (*p* < 0.1) increased (Figure 7).

## 3. Discussion

During this study, H_2_O_2_ was used to mitigate cyanobacterial populations in environmental samples from four water bodies. H_2_O_2_ almost always reduced the levels of cyanobacteria and increased the abundances of eukaryotic algae. Effects on non-photosynthetic prokaryotes were complex, as some bacteria were consistently inhibited by H_2_O_2_ while others were promoted. Collectively, these findings provide new insight into the complex manner in which H_2_O_2_ can alter the composition of microbial communities dominated by cyanobacteria.

### 3.1. Cyanobacteria vs. Eukaryotes

Cyanobacterial biomass and cell abundance were significantly reduced by H_2_O_2_ in a majority of experiments (91%, 10 of 11), and eukaryotic green and unicellular brown algae were significantly increased in 73% and 55% of experiments, and did not decline significantly in any experiment. The ability of H_2_O_2_ to significantly reduce concentrations of cyanobacteria relative to eukaryotic algae is consistent with previous observations [11,13,14]. Across all experiments, the fluorometric relative abundance of cyanobacteria was significantly reduced from 85 ± 8% to 29 ± 10% (*p* < 0.005). The trend was also reflected in microscopy, with significant cyanobacterial reductions and significant increases of eukaryotic algae in all experiments quantified. Differences in flow cytometry were more varied, with *Cyanobium* and eukaryotic algae counts each significantly lower than control in one experiment, and significantly higher in another. A lack of ascorbate peroxidases in cyanobacteria, which are common in eukaryotes, has been offered as one possible factor contributing to their greater sensitivity [20,21]. Additionally, photosystem II of cyanobacteria is not protected within a cell organelle, making it physically more susceptible to damage from H_2_O_2_ [22].

Despite the common reduction in cyanobacterial biomass in experiments, it was reduced to levels below detection limit in only two out of 11 experiments and reduced below 25 µg Chl*a* L^−1^ (NYSDEC guidance value for cyanobacterial blooms) in one other. The inability to completely eliminate cyanobacteria may result in their resurgence over a short period of time [11]. While microcystin was only quantified once in this study, whole-water concentrations slightly increased (0.6 to 0.9 µg L^−1^) in that experiment, a finding that slightly contrasts with prior studies [11] but is perhaps not completely surprising given the dynamics of cyanobacteria in that experiment. While total cyanobacterial biomass decreased in that experiment, it was one of the smallest declines in this study (33%). Further, while one microcystin producer, *Microcystis* [1], significantly declined in relative abundance during that experiment, another, *Planktothrix* [1], significantly increased and seemingly contributed to the slightly higher toxicity observed at the end of this experiment.

### 3.2. Comparing Cyanobacterial Genera

While cyanobacterial biomass and cell densities were reduced in a majority of experiments, the reductions varied between genera. Although cyanobacteria do not usually produce ascorbate peroxidases, they are capable of producing a suite of other anti-oxidant enzymes, including catalases, peroxidases, and peroxiredoxins [21]. While peroxiredoxins are ubiquitously present across cyanobacteria, other enzymes can vary between genera and strains, which may account for some of the variability in relative reductions observed [21].

*Microcystis* was the most ubiquitous of the cyanobacteria during this study and was significantly reduced in microscopic counts to levels less than control in all experiments, but was reduced below detection in only one. *Microcystis* was reduced to a significantly lower relative and absolute abundance compared to the unamended controls in two of the three experiments where sequencing was performed and was higher with H_2_O_2_ in the experiment where *Planktothrix* was dominant. In contrast to *Microcystis*, *Cyanobium* and *Cylindrospermopsis* were more resistant to H_2_O_2_. *Cyanobium* was the second most abundant cyanobacteria in Roth Pond, behind *Microcystis*, and cell concentrations were significantly lower relative to the control in one experiment, and significantly higher in another. The sequenced relative abundance of *Cyanobium* was significantly lower in one of the three experiments. A prior laboratory study found *Cyanobium* four-fold less sensitive to H_2_O_2_ than *Microcystis* [13]. *Cylindrospermopsis* concentrations in microscopy were detected in two experiments, and were significantly reduced in only one (Figure 4c). The sequenced relative abundance of *Cylindrospermopsis* in H_2_O_2_ treatments was higher than the control in all three experiments, including one where it was dominant over *Microcystis*, though it was still reduced in absolute abundance (Figure 4c and Figure 5).

The heightened sensitivity of *Microcystis* to H_2_O_2_ relative to other genera may be related to its deficient antioxidant systems as some strains of *Microcystis* lack typical cyanobacterial catalases [21]. Importantly, however, while *Microcystis* densities were significantly reduced by initial treatment with H_2_O_2_, it often remained one of the most abundant cyanobacterial genera in experiments where it dominated and was never fully eliminated. *Microcystis* commonly forms large globular colonies with cells embedded in and surrounded by polysaccharide mucous [23]. The extracellular polymeric substances of this mucous have strong H_2_O_2_ scavenging abilities and provide an antioxidant buffer for the cells within [24]. This additional protection may partly explain the perseverance of *Microcystis* during experiments. In addition, it has been shown that some strains of *Microcystis* incapable of microcystin synthesis have high levels of thioredoxin and peroxiredoxin, enzymes involved in H_2_O_2_ degradation [19] making these strains more likely to survive repeated H_2_O_2_ doses than toxic strains. Hence, the persistence of *Microcystis* may be partly facilitated by shifts among differing strains.

The relative H_2_O_2_ resistance of *Cylindrospermopsis* among cyanobacteria may be due, in part, to their ability to produce superoxide dismutase, catalase, and ascorbate peroxidase [25]. The ability to produce ascorbate peroxidase in *Cylindrospermopsis* is somewhat unique, as it was believed to be lacking in other cyanobacteria, and therefore contributes to the reduced effectiveness of peroxide against this genus [17,20]. In addition, *Cylindrospermopsis* can produce single cell akinetes which it uses to survive unfavorable conditions [26]. Akinete production may also account for differences between gene detection and microscopic counts where *Cylindrospermopsis* reduced was below microscopic detection, as the visible trichomes may have fragmented into akinetes or small, morphologically unidentifiable fragments [27]. It is unlikely the 16S rDNA of *Cylindrospermopsis* persisted after the cells were destroyed by H_2_O_2_, as DNA often degrades within 24 h in freshwater [28,29].

*Planktothrix* dominated the Lake Agawam experiment that was examined in detail and displayed the largest relative decline in sequenced abundance of any microbe, dropping from 94% to 17% in the H_2_O_2_ treatment. In the Roth Pond experiment, *Planktothrix* increased slightly in relative and absolute abundance of sequences but was below detection levels in microscopic counts. H_2_O_2_ has been shown to be effective in controlling *Planktothrix* in lakes and ponds at similar concentrations to those used here [11,30,31]. This experiment where *Planktothrix* declined by nearly 80% in relative abundance was also the only instance when the sequenced relative abundance of *Microcystis* among the cyanobacteria increased following treatment with H_2_O_2_, despite its absolute decline, supporting the hypothesis that *Planktothrix* is more sensitive to H_2_O_2_ than *Microcystis* [30,32], and in assessing all available literature [11,31,32], may be the most sensitive of the cyanobacterial genera to H_2_O_2._

The collective response of these experiments leads evidence to support an ‘open niche’ hypothesis [33]. with regard to the effects of H_2_O_2_ on cyanobacteria. While different cyanobacteria are likely differentially sensitive to H_2_O_2_, it also seems that the effects of H_2_O_2_ are somewhat conditional upon the original community composition. That is, in many cases the dominant cyanobacterial genus is most reduced by H_2_O_2_, perhaps by providing the most organic surface area for the H_2_O_2_ to react with, allowing genera at lower relative abundances to fill the niche left open by the formerly dominant genera.

### 3.3. Heterotrophic Bacteria

The net effect of H_2_O_2_ on bacteria during this study was inconclusive, as concentrations of heterotrophic bacteria were significantly lower than the control in one of three experiments, and significantly higher in another. In contrast, high throughput sequencing of 16S rDNA revealed that, beyond with changes in total bacterial densities, there were pronounced shifts within prokaryotic communities following treatment with H_2_O_2_. The prokaryote taxa identified here were categorized at the phylum level; some traits discussed below may not be indicative of all taxa within a phylum, especially for the very diverse *Proteobacteria,* but offer some insight to potential strategies employed.

Consistent with fluorometric and microscopic evaluations, the sequenced relative abundance of cyanobacteria made up an average 18 ± 9% of initial abundances and was significantly reduced in two of three experiments. The relative abundance of *Actinobacteria*, which made up an average 11 ± 3% of initial abundances, was likewise reduced in two of three experiments of experiments, demonstrating that strains of bacteria within this phylum are susceptible to H_2_O_2_. In lakes, *Actinobacteria* are small, thin walled, free-living ultramicrobia (< 0.1µm^3^) that are abundant in the epilimnion [34]. *Actinobacteria* are obligate aerobes [35], and densities decrease with decreasing oxygen levels and depth [34,36]. They are defense specialists that are relatively resistant to grazing, but have slow growth rates compared to other bacteria [34,37]. These slow growth rates may make them less likely to recover from initial populations declines induced by H_2_O_2_.

Planctomycetes and Verrucomicrobia, which have a close phylogenetic relationship [38], made up a smaller portion of the total bacterial community (< 5%) but also appeared susceptible to H_2_O_2,_ significantly decreasing in sequenced relative abundance in two of three experiments, and in all experiments, respectively. Verrucomicrobia are usually found throughout the water column and are associated with high-nutrient environments and algal blooms [34]. Prior mesocosm experiments have shown Verrucomicrobia to strongly increase in response to Microcystis degradation [39]. Planctomycetes similarly are capable of breaking down high-molecular weight organic compounds [40,41]. Both phyla are also capable of producing bifunctional catalase-peroxidases [42]. Peroxide resistance, benefitting from organic carbon, and declining Microcystis abundances should have all theoretically promoted these groups. The absence of such a response in their sequenced relative abundance may have been a function of a larger increase or lesser decline in other bacterial groups or a stronger negative effect of H_2_O_2_ relative to any benefit derived from algal organic matter.

*Proteobacteria* increased significantly in sequenced relative abundances in one of three experiments. *Proteobacteria* were one of the most abundant prokaryotes, making up 30 ± 8% of initial abundances. *Proteobacteria* have a relatively short generation time, allowing them to respond rapidly to changing conditions [34] and are copiotrophic, able to assimilate small organic acids as well as degrade complex organic compounds [34,40,41]. *Proteobacteria* also contain catalase-peroxidases and manganese catalases that protect them from oxidative stress [42].

The average initial relative abundance of *Bacteroidetes* was 27 ± 4%, and values were significantly higher following H_2_O_2_ exposure in two of three experiments. *Bacteroidetes* are usually particle associated and chemoorganotrophic [34], and are capable of degrading complex, high-molecular weight organic compounds [34,40,41]. *Bacteroidetes* become abundant when DOC or algae-derived DOC are high [34], and *Bacteroidetes* abundances often increase during the degradation of *Microcystis* blooms [39]. Their ability to benefit from H_2_O_2_ in two experiments and to be unaffected in a third may relate to their exploitation of DOC released by lysing cyanobacteria and/or their association with particles that might scavenge some H_2_O_2_ and thereby protect attached cells.

The findings presented here contrast slightly with Lin et al. (2018) who similarly performed mesocosm experiments and analyzed the 16S rDNA for bacterial community shifts in response to 8mg H_2_O_2_ L^−1^. While *Firmicutes* increased in sequenced relative abundance in that study, *Proteobacteria* and *Bacteroidetes* were reduced [32], whereas these two groups increased or were generally unchanged in the present study. Those results, however, emanated from a single experiment that was performed in winter in Dianchi Lake, China, where the temperature was 10 °C [32]. Moreover, changes in relative abundance can be complex as changes in relative abundance of any one group is also dependent on the response of other groups.

### 3.4. Comparison Among Ecosystems

The reactivity of H_2_O_2_ is influenced by its rate of decay, which can be affected by oxidation-reduction processes and the organic matter content of water bodies [43,44]. The nearest measure of organic matter concentration this study performed was algal biomass. Georgica Pond samples had the lowest total biomass with a concentration of 97 µg Chl*a* L^−1^, and a cyanobacterial biomass of 73 µg Chl*a* L^−1^, which was reduced 99.8% by H_2_O_2_. Lake Agawam was the next densest, with an average total biomass of 164 µg Chl*a* L^−^^1^, 138 µg Chl*a* L^−1^ of which was cyanobacteria, which was reduced by 53%. Mill Pond experiments had an average total biomass of 267 µg Chl*a* L^−1^, with an average cyanobacterial concentration of 241 µg Chl*a* L^−1^, which was reduced by 96%. Roth Pond was the densest and most mixed, with an average total biomass of 486 µg Chl*a* L^−1^, and an average cyanobacterial concentration of 140 µg Chl*a* L^−1^, which was reduced by 67%. Across these four systems, there was no significant relationship between effectiveness of H_2_O_2_ in reduction of cyanobacterial biomass and total biomass (Figure 8a), and, therefore, other factors may have had a greater influence on efficacy. For some individual systems, specifically Lake Agawam and Roth Pond, increasing levels of algal biomass were significantly correlated with decreasing reductions on cyanobacterial biomass during H_2_O_2_ treatments (Lake Agawam, *p* < 0.005, Figure 8b; Roth Pond, *p* < 0.001, Figure 8c), meaning H_2_O_2_ became less effective at controlling cyanobacteria as total algal biomass increased. This trend was not detected in water from Mill Pond or Georgica Pond. Regardless, the findings for Lake Agawam and Roth Pond samples suggest that the efficacy of H_2_O_2_ in controlling cyanobacteria can depending on the levels of total algal biomass, but that the relationship is partly conditional upon other factors that may differ independently across ecosystems including levels of total organic carbon in the water column and the inventory of organic matter within sediments.

Across all experiments, initial reductions in cyanobacterial biomass persisted during the four-to-seven-day incubations. Other long-term studies from lakes and wastewater stabilization ponds dominated by *Planktothrix* found the effects of H_2_O_2_ persisted for five to seven weeks [11,31], and in a mixed assemblage of *Microcystis* and *Planktothrix* persisted for three weeks [17]. Longer observations would be required to determine the ability of the cyanobacteria to recover and the time frame within which such a recovery would occur.

## 4. Conclusions

This study demonstrated that H_2_O_2_ administered at a moderate dose (4 mg L^−1^) consistently inhibits cyanobacteria and promotes the growth of eukaryotic algae, primarily green algae. H_2_O_2_ significantly reduced heterotrophic bacterial densities, with the phylum *Actinobacteria* most consistently reduced. Other bacterial phyla also declined, but were relatively less impacted, potentially due to their antioxidant enzymes and recovery fueled by use of cyanobacterial-derived organic matter, with some phyla increasing in relative abundance following the addition of H_2_O_2_. H_2_O_2_ did not successfully reduce cyanobacteria levels below detection or guidance levels in a majority of experiments at the concentrations used here (4 mg L^−1^). While the elimination of cyanobacteria might have been achieved with higher doses of H_2_O_2_, prior research has suggested that such levels may cause collateral damage on non-target organisms [11,12]. The use of bottle experiments here, however, likely maximized contact between H_2_O_2_ and the plankton community, while minimizing scavenging by sediments. Future, larger scale and whole ecosystem experiments may provide deeper insight with regard the true effect and effectiveness of H_2_O_2_ in altering plankton community structure and mitigating cyanobacterial blooms.

## 5. Materials and Methods

Bottle incubation experiments were performed within four study systems: Georgica Pond (latitude, longitude = 40.938671, −72.230216), Mill Pond (40.914848, −72.358037), Lake Agawam (40.875074, −72.391942), and Roth Pond (40.911734, −73.123772). Lake Agawam, Mill Pond, and Georgica Pond are shallow (2–3 m) natural freshwater (Lake Agawam, Mill Pond) or brackish (Georgica Pond; salinity 0–20 PSU) water bodies located on Long Island’s south shore and are 0.24 km^2^, 0.49 km^2^, and 1.17 km^2^, respectively, and known to experience repeated cyanobacterial blooms [45,46]. Roth Pond is a man-made, 3 × 10^−3^ km^2^, 1 m deep water body on the campus of Stony Brook University, in Stony Brook, NY, USA.

This study presents 11 experiments performed in two rounds (Table 1). The first round of experiments consisted of three experiments from Lake Agawam, three experiments from Mill Pond, one experiment from Roth Pond, and one experiment from Georgica Pond that were evaluated for changes in phytoplankton communities in response to H_2_O_2_ using a bbe Moldaenke (Kiel, Germany) Fluoroprobe. The second round of experiments were performed at Lake Agawam, Mill Pond, and Roth Pond and involved in-depth and detailed investigations of the prokaryotic and eukaryotic plankton communities’ responses to H_2_O_2_ using fluorometry, microscopy, flow cytometry, and high-throughput amplicon sequencing.

Surface water was collected using 20 L carboys from each site experiencing a cyanobacterial bloom defined as > 25 µg Chl*a* L^−1^ from cyanobacteria as quantified on a bbe Fluoroprobe according to the (NYSDEC; details below). Bloom water was transferred from the carboys into 4-L polycarbonate bottles using acid-washed Tygon tubing placed at the bottom of the bottle to reduce bubbling and disturbance of the plankton. Each bottle experiment consisted of two treatments, with three replicate bottles each: an unamended control, and 4 mg H_2_O_2_ L^−1^ treatment achieved via the addition of a 3% w/v H_2_O_2_ solution. These H_2_O_2_ levels have been previously shown to reduce levels of *Planktothrix* but not eukaryotes in a matter of days in European lakes [11,12]. Bottles were incubated for four-to-seven days under ambient light and temperature conditions in an outdoor flow-through table, constantly flushed with water from Old Fort Pond, Southampton, NY, which maintains water temperatures comparable to the shallow lakes and ponds studied here (Figure 9). Initial and final timepoint samples were obtained and preserved for microscopy (5% Lugol’s iodine), flow cytometry (10% buffered formalin stored at −80 °C), and the extraction of DNA (50 mL onto a 0.2 µm, 47 mm Isopore polycarbonate filter frozen at −80 °C). Lugol’s iodine preserved samples were analyzed using a Sedgewick Rafter slide to quantify cyanobacteria at the genus level as well as eukaryotic algae that were broadly categorized as unicellular green algae (chlorophytes) or diatoms. Formalin-preserved samples were used to quantify the abundances of phycocyanin-containing pico-cyanobacteria, pico- and nano-eukaryotic phytoplankton, and SYBR Green I-stained heterotrophic bacteria on a CytoFLEX flow cytometer (Beckman Coulter, Indianapolis, IN, USA) based on fluorescence patterns and particle size [47]. Pico-cyanobacteria are identified here as *Cyanobium* based on high-throughput sequencing identification results. Fluorescence measurements were made on initial and final live samples on a BBE Fluoroprobe which categorizes algal biomass (Chl*a*) based on the fluorescence signatures of green algae, cyanobacteria, and unicellular brown algae (diatoms, dinoflagellates, raphidophytes) [48]. Whole-water samples (1 mL) were collected and frozen for one experiment for analysis of the cyanobacterial toxin, microcystin, via an Abraxis Microcystins ELISA assay which included lysing cells for a total concentration [49]. Differences between treatments and controls for each parameter measured in the experiments were assessed via a one-way ANOVA. Values below detection were entered as zero.

### DNA Extraction, Sequencing, and Analysis

DNA barcoding analysis was performed to assess changes in community composition of cyanobacteria and other prokaryotes. Samples were initially heated in a water bath to 65° for 10 min to aid in lysing, and extractions were performed using a Qiagen DNeasy^®^ PowerWater^®^ Kit. Double-stranded DNA was quantified on a Qubit^®^ fluorometer using a dsDNA BR Assay kit. Aliquots were normalized to an equal quantity of DNA, and sent to Molecular Research Laboratories (Shallowater, TX, USA) for amplicon sequencing. Paired-end sequencing was performed on an Illumina MiSeq (2 × 300bp) following the manufacturer’s guidelines. The 16S rRNA gene V4 variable region (~252bp) was amplified using universal primers 515F: 5’-GTG YCA GCM GCC GCG GTAA-3′ [50] and 806RB: 5´-GGA CTA CNV GGG TWT CTA AT-3′ [51]. For each sample, an identifying barcode was placed on the forward primer and a 30 cycle PCR using the HotStarTaq Plus Master Mix Kit (Qiagen, Valencia, CA, USA) was performed. The following PCR conditions were used: 94 °C for 3 min, followed by 28 cycles of 94 °C for 30 s, 53 °C for 40 s and 72 °C for 1 min, and a final elongation step at 72 °C for 5 min. Samples were purified using calibrated Ampure XP beads and subsequently used to prepare an Illumina DNA library.

Sequence data was processed using the Quantitative Insights into Microbial Ecology QIIME 1 (v1.9.1) and QIIME 2 (v2019.1.0) following the “Moving Pictures” pipeline (QIIME, http://qiime.org; [52]). Raw sequences were depleted of sample barcodes in QIIME 1. Paired-end reads were demultiplexed in QIIME 2 using the DEMUX plugin, and were depleted of primers using the Cutadapt Plugin. The library was filtered for chimeric sequences, denoised and dereplicated using the DADA2 plugin. A naïve Bayes classifier was trained using the SILVA rRNA (16S SSU) v132 reference database at 99% similarity, and was used with the q2-feature-classifier and classify-sklearn plugins to assign taxonomies. The dataset was filtered to remove mitochondria and chloroplast features. Prokaryotic OTUs were examined at the phylum level and were expressed as relative abundance of each. Cyanobacteria OTUs were then examined at the genus level, also expressed as relative abundance. For each set, prokaryote OTUs and cyanobacteria OTUs, relative abundances exceeding 5% were compared, and the remaining were grouped as “other”. Statistical analysis of diversity, as well as shifts in relative abundance were performed in QIIME 2. Absolute abundances of cyanobacterial genera were estimated by multiplying the Fluoroprobe quantified levels of cyanobacterial fluorescence by the relative abundance of each genera exceeding 5% of the total OTU reads. Raw sequencing data was deposited to the NCBI Sequence Read Archive (SRA) under accession number PRJNA642309.

## Figures and Tables

**Figure 1 toxins-12-00428-f001:**
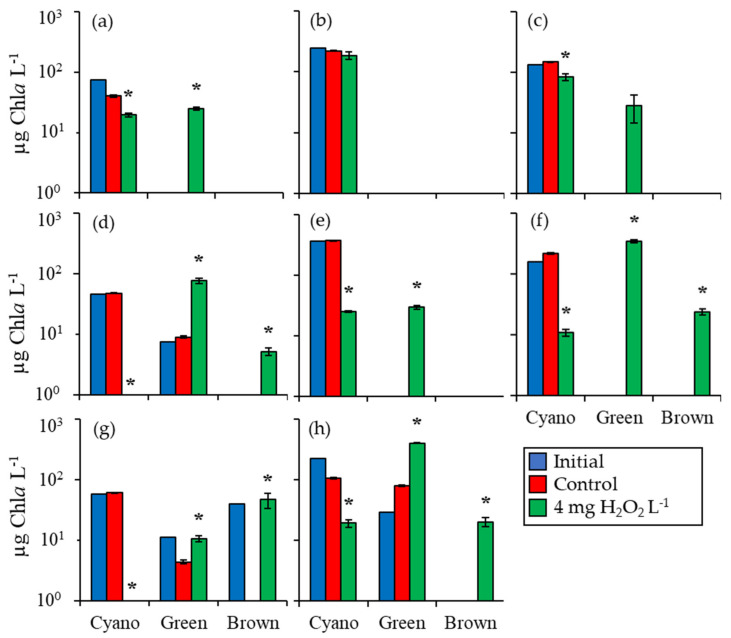
Fluoroprobe biomass measurements for bottle experiments from Lake Agawam (**a**) 7/21/16, (**b**) 10/20/16, (**c**) 6/9/17; Mill Pond (**d**) 7/21/16, (**e**) 10/20/16, (**f**) 6/30/17; (**g**) Georgica Pond 7/21/16; and (**h**) Roth Pond 6/9/17. Asterisks show significant changes (*p* < 0.05) in treatments relative to control. Error bars show standard error.

**Figure 2 toxins-12-00428-f002:**
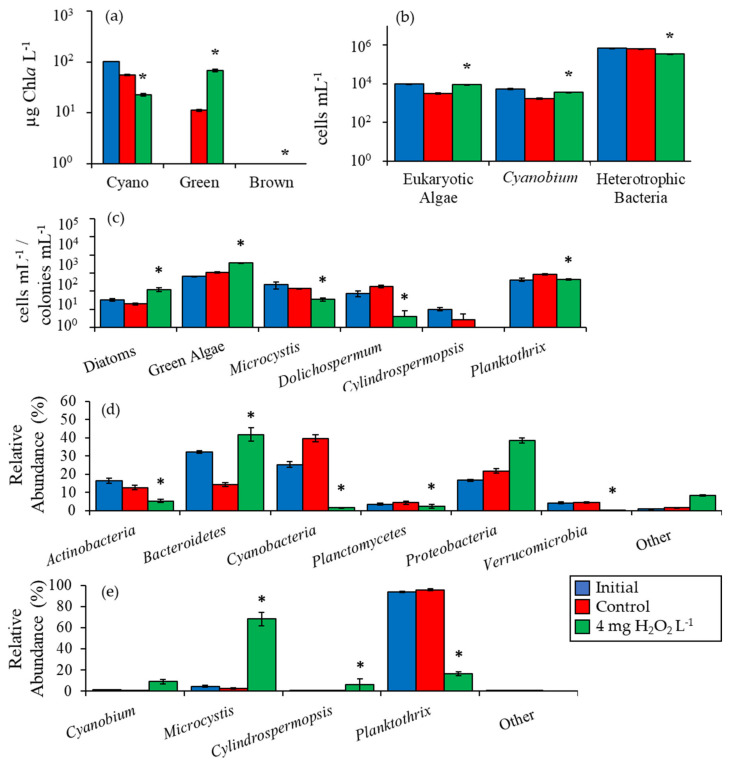
(**a**) Fluoroprobe biomass, (**b**) flow cytometry, (**c**) microscopy, (**d**) phylum level relative abundance, and (**e**) genus level cyanobacteria relative abundance for Lake Agawam experiment, 10/3/16. Asterisks show significant changes (*p* < 0.05) in treatments relative to control. Error bars show standard error.

**Figure 3 toxins-12-00428-f003:**
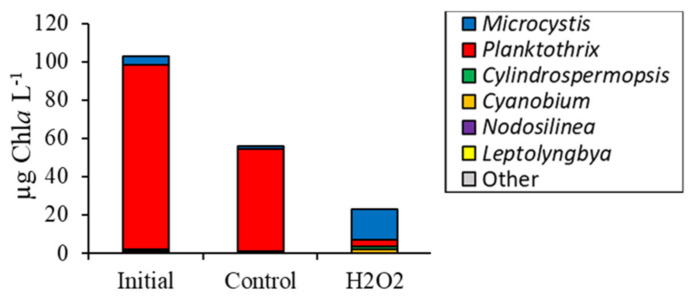
Absolute abundance (relative abundance multiplied by biomass) of cyanobacteria for Lake Agawam experiment, 10/3/16.

**Figure 4 toxins-12-00428-f004:**
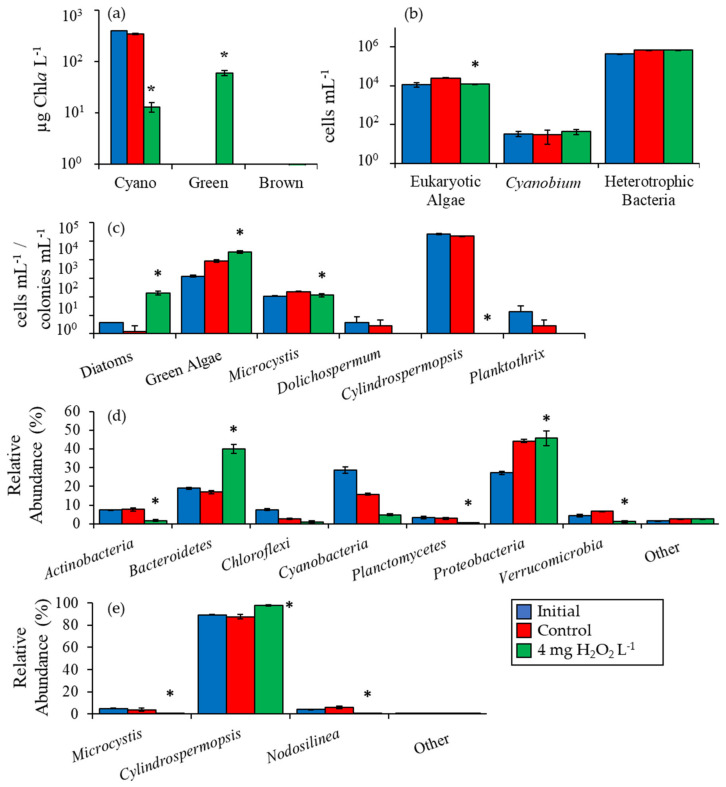
(**a**) Fluoroprobe biomass, (**b**) flow cytometry, (**c**) microscopy, (**d**) phylum level relative abundance, and (**e**) genus level cyanobacteria relative abundance for Mill Pond experiment, 10/3/16. Asterisks show significant changes (*p* < 0.05) in treatments relative to control. Error bars show standard error.

**Figure 5 toxins-12-00428-f005:**
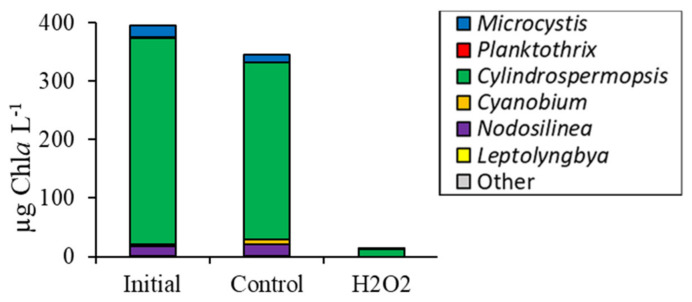
Absolute abundance (relative abundance multiplied by biomass) of cyanobacteria for Mill Pond experiment 10/3/16.

**Figure 6 toxins-12-00428-f006:**
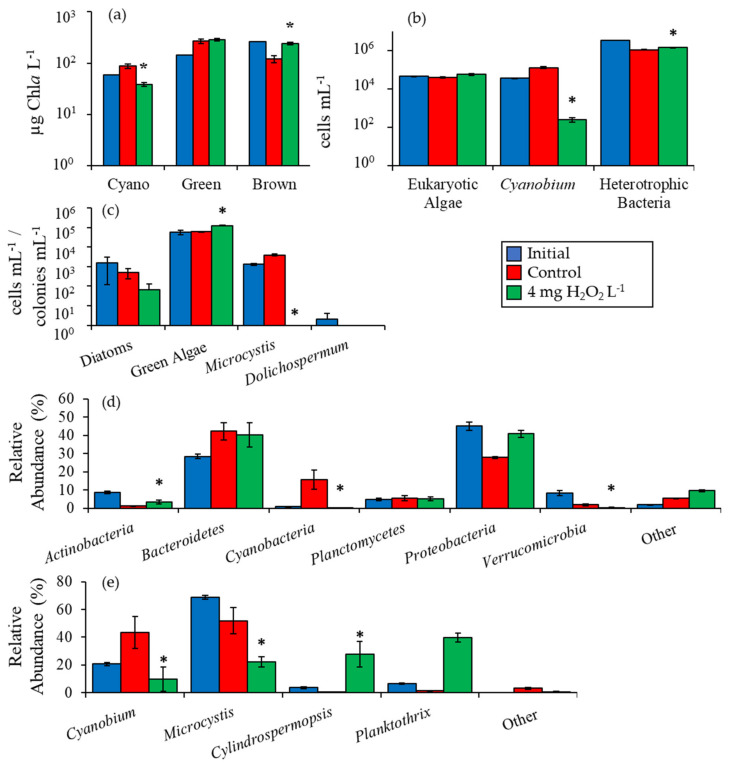
(**a**) Fluoroprobe biomass, (**b**) flow cytometry, (**c**) microscopy, (**d**) phylum level relative abundance, and (**e**) genus level cyanobacteria relative abundance for Roth Pond experiment, 6/30/17. Asterisks show significant changes (*p* < 0.05) in treatments relative to control. Error bars show standard error.

**Figure 7 toxins-12-00428-f007:**
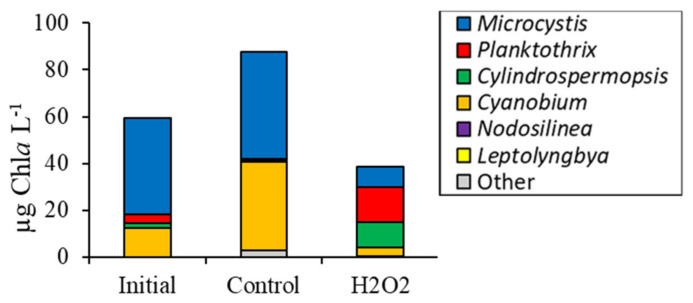
Absolute abundance (relative abundance multiplied by biomass) of cyanobacteria for Roth Pond experiment, 6/30/17.

**Figure 8 toxins-12-00428-f008:**
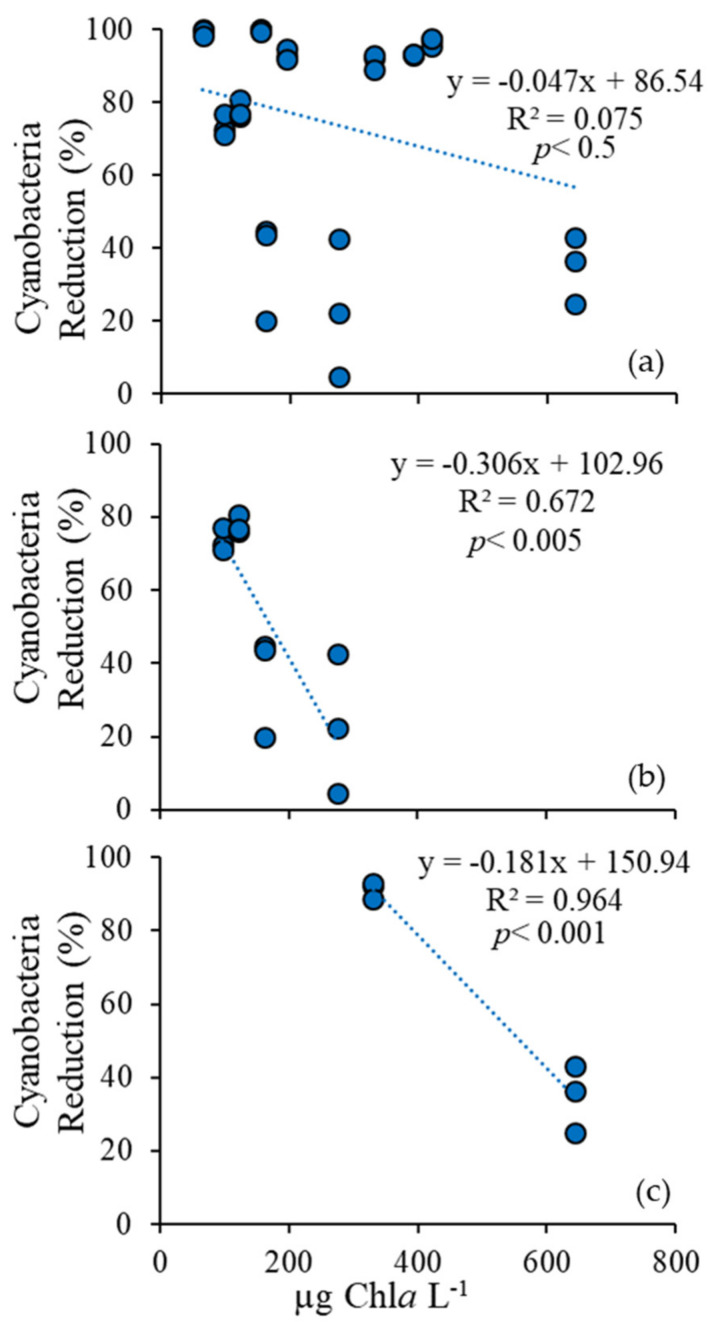
Regression plots for percent cyanobacterial reduction per initial unit biomass (µg Chl*a* L^−1^) following the addition of 4 mg L^−1^ H_2_O_2_ for all experiments (**a**), Lake Agawam experiments (**b**), and Roth Pond experiments (**c**).

**Figure 9 toxins-12-00428-f009:**
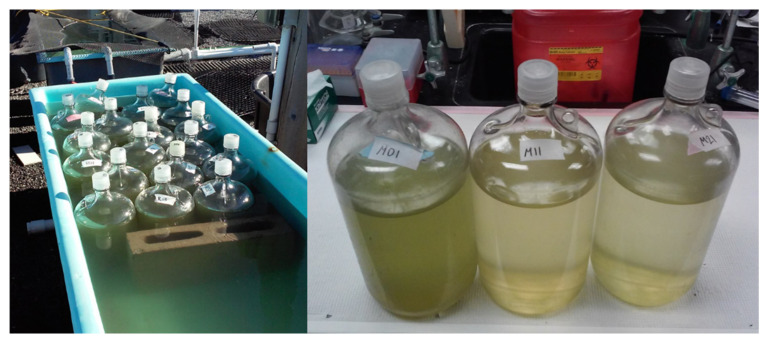
Outdoor flow-through table, and 4 L bottles used for incubation experiments. Example of pigment change between control and H_2_O_2_ treatments.

**Table 1 toxins-12-00428-t001:** Overview of experiments and analysis performed for each waterbody.

Waterbody	Dates	Analysis
Lake Agawam	7/21/16 10/20/16 6/9/17	Fluorescence only
	10/3/16	Full^1^
Mill Pond	7/21/16 10/20/16 6/30/17	Fluorescence only
	10/3/16	Full^1^
Georgica Pond	7/21/16	Fluorescence only
Roth Pond	6/9/17	Fluorescence only
	6/30/17	Full^1^

^1^ Fluorescence, microscopy, flow cytometry, DNA sequencing.

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
