# Peer review of "The Efficacy of Hydrogen Peroxide in Mitigating Cyanobacterial Blooms and Altering Microbial Communities across Four Lakes in NY, USA"

_toxins, 2020, doi:10.3390/toxins12070428_

Round 1
Reviewer 1 Report
The manuscript is very well presented and contains publishable data. However, minor modifications are necessary before publication:
-Line 71: Microcystis, Planktothrix, and Dolichospermum should be in italic. Check throughout the manuscript for cyanobacterial genera that should be in italic (lines: 80, 91, 103).
-Line 136: Dolichospermum did not form colonies but in solitary filaments.
- Line 138: the same remark for Planktothrix.
- Line 181: the same remark for Cylindrospermpsis.
Comment:
- Line 263: The authors reported that microcystins slightly increased after treatment with H2O2. In the material method section there is no details how samples used for microcystins analysis are treated. The authors indicated that samples (1 ml) were collected and frozen for analysis of microcystins. It's not clear if dissolved or particulate fractions of toxins were tested. H2O2 can destroy cyanobacterial cells and therefore, realising of microcystins which explains the slightly increased of MCs after treatment with H2O2.
Author Response
-Lines 71, 80, 91, 103: Fixed italicization of genus names. Also fixed other italics; 351-362.
-Line 136, 138, 181: Changed units for Dolichospermum, Planktothrix, and Cylindrospermpsis counts from "colonies mL-1" to "chains mL-1".
- Line 263, 474, 476-477; microcystin analysis: Inserted text to indicated samples were lysed for a total concentration.
Thank you for the feedback. Stay safe.
Reviewer 2 Report
In this work, the authors examine the use hydrogen peroxide to reduce the presence of cyanobacteria in environmental samples. In most cases they saw the overall reduction of cyanobacteria in the samples, while the total algal diversity was significantly affected. The work is interesting and the manuscript is well-written, therefore I would like to recommend it for publication after the authors have addressed the comments below.
Major comments
- The manuscripts through the introduction, results, and discussion sections reads as if the authors performed the experiments in field trials. This is obviously not true, as the experiments took place in the lab using environmental samples. I think this misconception should be clarified in each section to better reflect the scope of the study
- In line 269 the authors make a remark about observed toxicity, while they provide no data. If the authors have studied the toxicity (of samples?) can they add the data and update the methods section?
- Expanding on the previous comment, the reduction of overall cyanobacterial content doesn't necessarily mean safer water. The authors show that the abundance of some microcystin-producing strains increases, and we don't know how the H2O2 stress might affect the cyanotoxin production. It would have been interesting if the authors measured cyanotoxin presence and concentration, though i think they should add a paragraph in the discussion or conclusions elaborating on this topic
Minor comments
- A picture of the experimental setup would greatly help the reader
- Can the authors elaborate on the feasibility of using H2O2 as conservation strategy?
Author Response
- Added text (lines 63-64, 70, 119, 169, 242-243, 391, 405-406), to clarify nature of environmental sample, and not in-pond experiments.
- Line 269; microcystin: Method lines 474-477. Concentration mentioned in line 266. I added data to results section to help clarify (lines 216-218)
- Added image of bottles and table to methods section. (lines 474-476)
- Cyanotoxins: We unfortunately don't have more toxin data for these experiments, and I don't want to draw too much from a singular result. It will be a focus in future research.